# Ecotoxicological Analysis of Emerging Contaminants from Wastewater Discharges in the Coastal Zone of Cihuatlán (Jalisco, Mexico)

**Miguel Ángel Arguello-Pérez [1], Jorge Alberto Mendoza-Pérez [2,\*], Adrián Tintos-Gómez [1,3], Eduardo Ramírez-Ayala [1], Enrique Godínez-Domínguez [1] and Francisco de Asís Silva-Bátiz [1]**

[1]  Department of Studies for the Sustainable Development of the Coastal Zones, University of Guadalajara, Jalisco 48980, Mexico
[2]  Department of Engineering in Environmental Systems, Laboratory of Clean Technologies, Development of Environmental Processes and Green Engineering, National School of Biological Sciences of National Polytechnic Institute, Mexico 07738, Mexico
[3]  Faculty of Marine Sciences, University of Colima, Colima 28860, Mexico
\*   Correspondence: jmendozap@ipn.mx

**Abstract:** This research provides a baseline tool to detect, predict and scientifically evaluate the toxic environmental impact generated by chemical substances that are categorized as emerging contaminants (ECs) with endocrine disruptive action. The present study was carried out in five effluents of wastewater produced by urban and rural settlements of the coastal zone of Cihuatlan, Jalisco. Five compounds, considered ECs and that act as endocrine disruptors, were analyzed: Diclofenac, Ibuprofen, Ketorolac, Pentachlorophenol (PCP), and Estradiol. The toxicity level (TEQ) of the ECs is estimated by a Quantitative Structure-Activity Relationship (QSAR) analysis, evaluating their concentration and assessing the risk involved in the incorporation of each one into the environment. The presence of the ECs was confirmed in all the studied sites. It was attested that the concentrations of pollutants Diclofenac, Ibuprofen, Ketorolac, and Pentachlorophenol were within the toxic range, whereas the compound Estradiol was found in concentrations that represent a high toxicity in the same effluents. This research recognizes that the analysis of the physicochemical properties of substances allows for predicting whether a contaminant is likely to act and persist in the environment and, in turn, bioaccumulate in organisms.

**Keywords:** ecotoxicology; ECs; endocrine disruptors; residual effluents; QSAR analysis; toxic equivalents

## 1. Introduction

The high levels of industrialization and urbanization along river basins have become a significant threat to coastal and estuarine ecosystems. These anthropogenic activities have resulted in increasing pollution load in various environmental compartments within the coastal ecosystem [1].

Different compartments of the coastal ecosystem are regarded as a sink for various types of chemical pollutants. Chemical contaminants, particularly emerging organic pollutants, have been detected in coastal zones at trace concentrations, which are still harmful enough concentrations to compromise the ecosystems [2]. It has been established that emerging contaminants (ECs) are probably the pollutants that cause the greatest concern, since their consumption is estimated in tons per year, and many of the most commonly used ones are employed in similar quantities to that of pesticides [3].

ECs are defined as previously unknown or unrecognized pollutants whose presence in the environment is not necessarily new, but concern about their possible consequences is now known [4]. Their study is one of the top priority research lines of the main organizations dedicated to the

protection of public and environmental health, such as the World Health Organization (WHO), the Environmental Protection Agency (EPA), or the European Commission. Emerging contaminants are compounds of which relatively little or nothing is known about their presence and impact on the different environmental compartments, which directly translates into a lack of regulations and a limited availability of methods for their analysis [5]. These compounds usually go undetected in the environment [3] and may occur naturally or synthetically [6].

Another particularity of these compounds is that, due to their high production, consumption, and their continuous introduction into the environment, they do not need to be persistent to cause significant negative effects by altering the endocrine system and blocking or disturbing hormonal functions, affecting the health of both human and animal species even when they are found in low concentrations [7].

The emerging contaminants can be found in many commonly used products, such as drugs, resins, plastics, pesticides, detergents, cosmetics, fragrances, etc. These compounds have the property of altering the hormonal balance of the endocrine system of an organism [8]. This alteration can be generated by blocking the hormonal action through competing with the hormone receptor, impersonating or mimicking the endogenous hormones, or by increasing or decreasing the levels of hormonal activity [9]. Since hormones are involved in the control of reproduction, sexual differentiation, organ coordination, brain organization, and metabolism, among others, such imbalance of the endocrine system may have a neurological and/or reproductive consequence in living organisms, representing a specific danger during the gestation phase and the initial stages of life [10].

The persistence of some EC compounds (such as Pentachlorophenol, cyclophosphamide, Ibuprofen, sulfamethoxazole, clofibric acid, among others) in the environment may span for more than a year, so they can progressively accumulate, reaching biologically active levels [11]. Due to the harmful effects of ECs, it is necessary to increase the knowledge about the origin, transformation, and effects of this new generation of pollutants, to propose treatment mechanisms in order to guarantee an ideal environmental quality and without causing adverse effects to organisms [12].

The investigation of the ecotoxic effect of a wastewater provides more solidity to the evaluation of environmental quality [13]. The most significant aspects are: The variation of the composition of the effluents, their identification as sources of contamination, the affectation of the optimal operation of biological treatment processes and the toxicity assessments [14].

*Ecotoxicology of Emerging Contaminants*

Ecotoxicology is a tool that serves for the evaluation of environmental quality, and it is considered to be the result of identifying both the disturbances and the impacts on an ecosystem or the environment generated by toxic pollutants. Since organisms are directly exposed to the combined effects of ecotoxicity, they are used as primary indicators of environmental health [15,16]. The use of bioassays is very important to determine the effects of ECs [17]. These studies establish the quality criteria for the protection of aquatic life, which is subsequently used to determine the environmental quality standards for each chemical compound [18]. The most commonly used toxicity parameters are the lethal concentration ($LC_{50}$), the effective concentration ($EC_{50}$) and the inhibition concentration ($IC_{50}$).

One way to evaluate toxicological effects is through methods that can relate structure, molecular properties, and the biological activities of the chemical compounds in the contaminants. The US-EPA proposed the calculation of the toxicity equivalence factor (TEF), which is a methodology used to evaluate the toxicity and risk of various substances [19]. Toxic equivalents (TEQ) are an estimated parameter that relate the toxicity of a compound with a reference component or value calculated by a bioassay [20]. The existence of numerical descriptors of the molecular structure (such as hydrophobicity, steric properties, molecular shape and topology, the degree of branching, molecular connectivity, etc.) and the availability of numerical parameters ($LC_{50}$, $EC_{50}$, $IC_{50}$, etc.) to measure biological activity, make it possible to apply computational methods to search for quantitative relationships between structure and activity, known as the QSAR model [21]. It shows that the

distribution, bioaccumulation, and biomagnification of these pollutants is conditioned by their physicochemical properties. Using these parameters, an estimate of the toxicity generated by a variable source can be found, that is, a weighting of how toxic an emission or discharge is [22].

The aim of this research is to contribute to the state of knowledge in the international scientific literature regarding environmental quality indicators, by generating an environmental toxicity index through an ecotoxicological analysis that evaluates the presence of emerging contaminants in wastewater effluents, their influence on the medium, and the effects on indicator organisms, in addition to the potential risks posed by the presence of these pollutants in the environment, in order to generate tools to introduce corrective measures that can contribute to the protection of water sources and the regulation of pollution in coastal zones by new generation compounds.

## 2. Materials and Methods

The present study includes the determination of the toxicity level of emerging contaminants by a Quantitative Structure-Activity Relationship (QSAR) analysis, which evaluates the EC's concentration and assesses the risk involved in the incorporation of them into the environment. Five ECs compounds [12,23] that act as endocrine disruptors were analyzed: Diclofenac, Ibuprofen, Ketorolac, Pentachlorophenol (PCP), and Estradiol.

### 2.1. Study Area

The study was carried out in five sites where wastewater is discharged on the coastal zone located in the municipality of Cihuatlán, Jalisco (Figure 1).

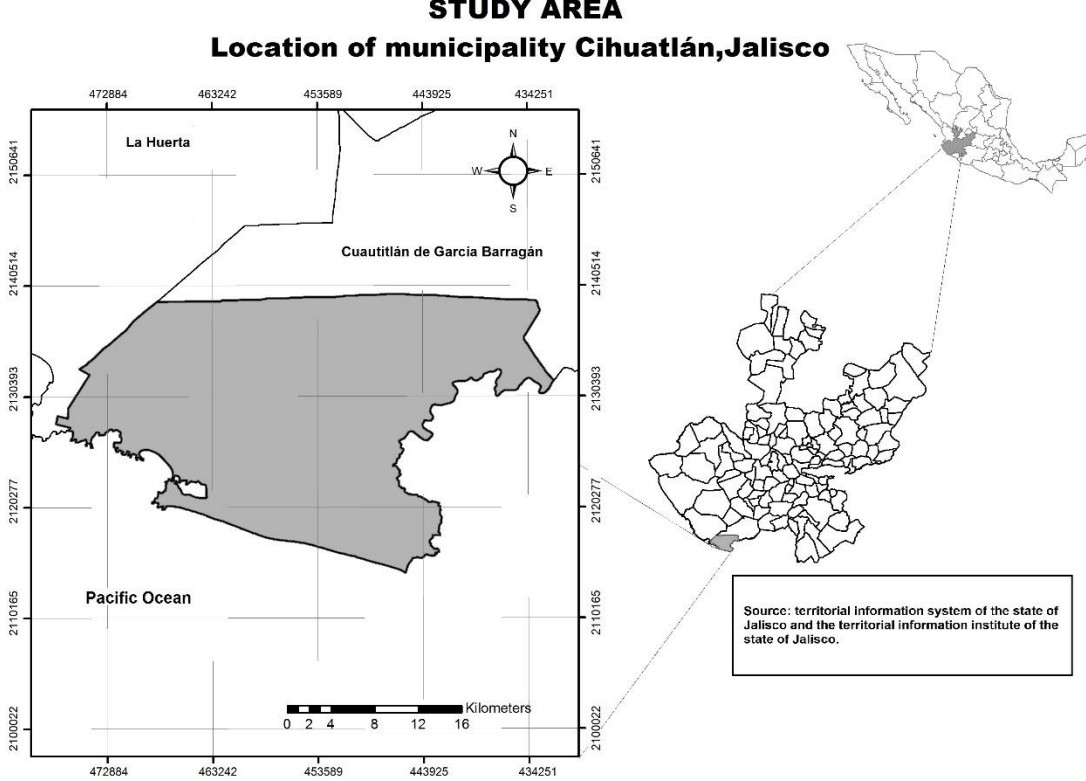

**Figure 1.** Location of the study area (municipality of Cihuatlán).

### 2.2. Water Sampling and Analysis

Water samples were collected in each of the five identified discharges (Figures 2–6).
The five residuals effluents studied were named as follows: "A", "D", "J", "L" and "S".

"A" effluent (Figure 2) catches and transports the wastewater discharged by two settlements of 230 and 600 inhabitants [24]. This effluent carries mainly agricultural wastewater that is discharged without any treatment onto nearby lands.

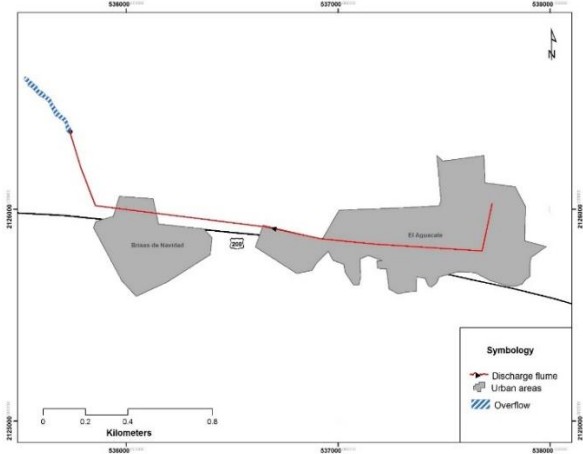

**Figure 2.** Location of "A" effluent.

"D" effluent (Figure 3) gathers and transports urban wastewater produced in the largest municipal settlement (32,000 inhabitants). It discharges without any treatment directly into two coastal lagoons [24], one of them is included in the list of Ramsar sites due to its ecological importance [25].

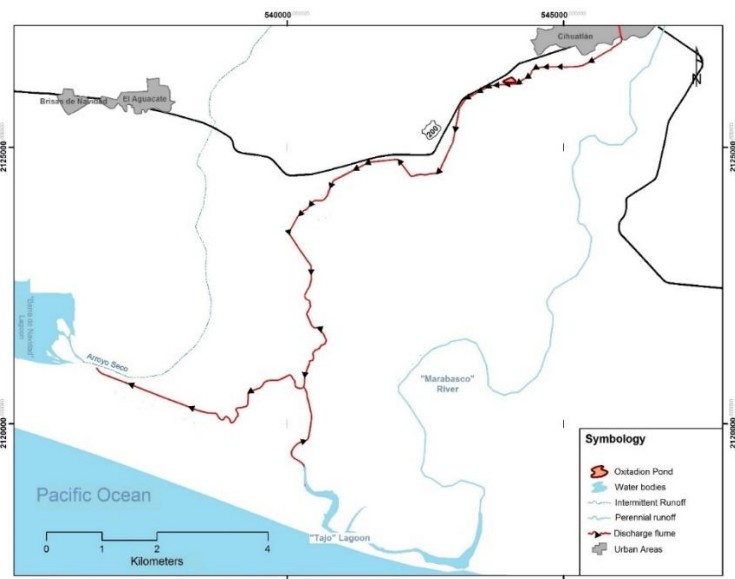

**Figure 3.** Location of "D" effluent.

"J" effluent (Figure 4) catches the urban wastewater produced by two settlements of 15,000 and 3200 inhabitants [24] and discharges without any treatment directly into a coastal lagoon.

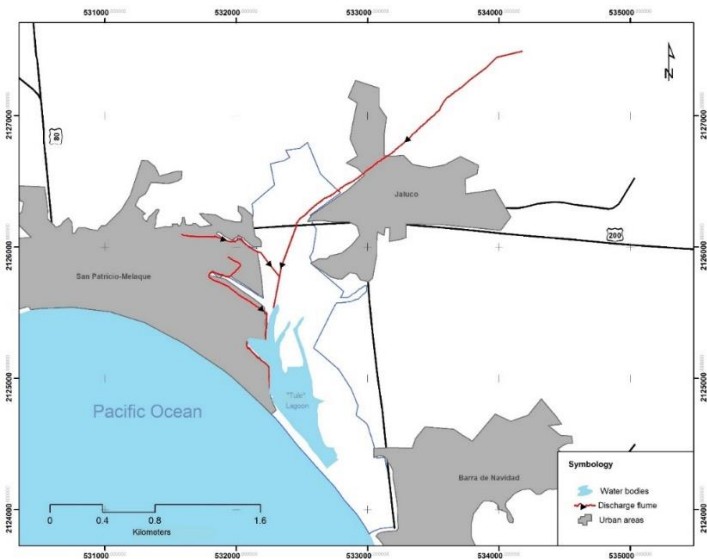

**Figure 4.** Location of "J" effluent.

"L" effluent (Figure 5) transports the wastewater produced by a settlement of 4320 inhabitants [24] and discharges into an overloaded oxidation pond as a final disposal site.

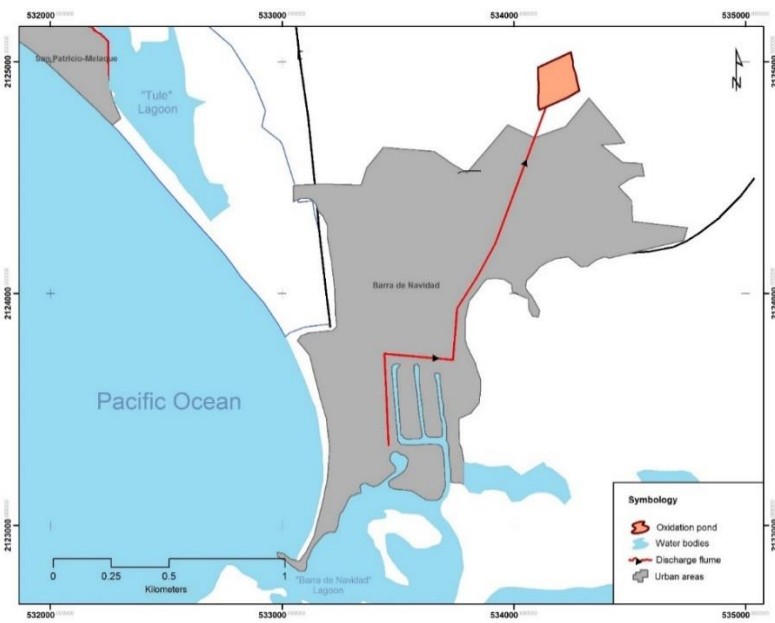

**Figure 5.** Location of "L" effluent.

"S" effluent (Figure 6) transports the wastewater produced by two settlements of 15,000 and 3150 inhabitants [24], and discharges into a non-operational wastewater treatment plant, and as final disposal site, land.

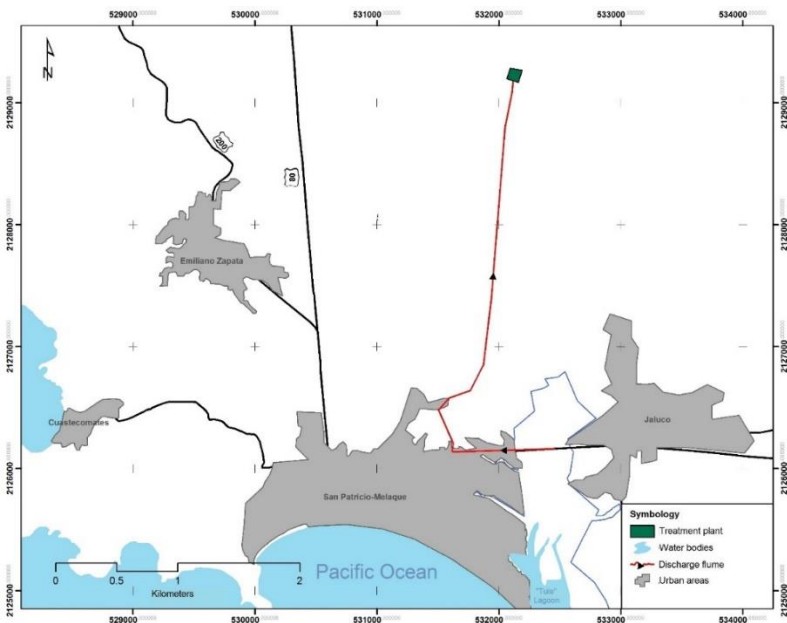

**Figure 6.** Location of "S" effluent.

The collection of water samples was carried out in accordance to the methodology described in the Mexican technical standard NMX-AA- 003 "Water analysis—residual water sampling". Samples were collected during the rainy season of the year 2017 [26] in a seven-day period at four-hour intervals.

The effluents were physically and chemically characterized according to the Mexican Official Standard NOM 001-SEMARNAT-1996—"Maximum permissible limits of pollutants in wastewater discharges into national waters and goods".

*2.3. Sample Processing*

The samples collected were processed according to method 525.2 "Determination of organic compounds by gas chromatography/mass spectrometry" of the US-EPA (1995).

*2.4. Analysis of ECs*

The UHPLC/MS/MS technique [27] was the method used to characterize contaminants as analytes.

Equipment characteristics: UHPLC 1290 Infinity II coupled to QTOF 6545 (Agilent Technologies). For data acquisition: UHPLC: MassHunter LC/MS Data Acquisition version B.06.01. For the mass analysis: MassHunter LC/MS Qualitative Analysis version B.07.00.

Chromatographic conditions: Mobile phase A: 0.1% formic acid, mobile phase B: methanol/formic acid 0.1%. Mobile phase C: Acetonitrile/formic acid 0.1%. Gradient: 0.0 min 50:50 (A:B v/v), 3.0 min 20:50:30 (A:B:C v/v/v), 10.0 min 20:50:30 (A:B:C v/v/v). Post time (equilibration time): 3.0 min. Injection volume: 20 µL. Flow: 0.3 mL/min. Run time: 10.0 min. Column: Zorbax Eclipse Plus C18 2.1 × 50 mm 1.8 µm. Column temperature 45.0 ± 0.5 °C. MS Analysis: Polarity: Positive. Source: Chemical ionization at atmospheric pressure (APCI). Range of 100 to 1000 m/z, Gas temperature: 250 °C. Vaporizer: 220 °C Gas flow: 5 L/min. Nebulizer: 60 psig. VCap: 3500 V. Crown: 4. Fragmentor: 110 V. Skimmer: 65 V.

Water samples were extracted with dichloromethane at HPLC chromatographic grade under reflux for 4 hours and then sonicated for 1 min/with 2 min pauses at 120 Hz for half an hour. The medium was concentrated with SPE C18 columns to fractions of 10 mL which were taken to 1 mL with rotary evaporator and injected to the equipment [28].

The quantification of these compounds was estimated using the MRM mode with a calibration curve and the identification of the compounds was carried out through mass spectrum comparison with

MERCK®analytical standards: Diclofenac sodium salt (329770242), Ibuprofen sodium salt (329815360), Ketorolac tris salt (24278503), Pentachlorophenol sodium salt (57652641), Estradiol (329749749).

Equipment detection limit: 0.02 ng/L. Limit of quantification of the method: 0.5 ng/L. Recovery rate: (91%, 95%) ± 1.5%. Matrix effect: (−9%, −3%) ± 0.5%. R2: (0.992, 0.998). Internal standards were used for each one with 10 ng/mL for the determinations of %Rec and −%ME.

To determine the toxicity of the contaminants, a quantitative analysis of the relationship between the structure and the chemical activity of the studied compounds was performed.

### 2.5. Quantitative Analysis of The Structure-Activity Relationship (QSAR) of ECs

The molecules of the ECs were plotted using GaussView®software (ver.5.0) from Gaussian Inc. ®, and optimized with the HartreeFock method (Figure 7) using base 6 calculation −31 g. Following the molecular optimization, descriptors were calculated combining information from the surface area of the molecule and its partial load, based on a series of weightings of the atom. Total energy, Bipolar Moment, octanol/water partition coefficient, obtained the energy values of the HOMO and LUMO boundary orbitals (highest occupied molecular orbital and lowest unoccupied molecular orbital, respectively) [21,29].

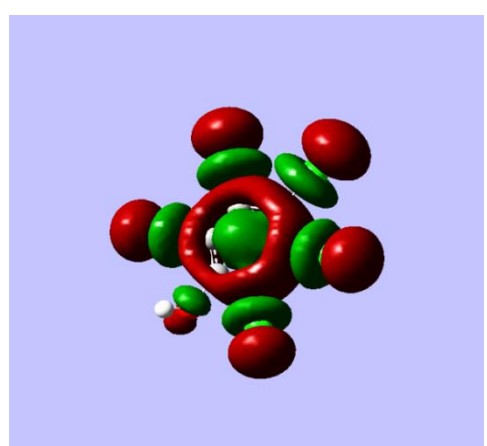

**Figure 7.** Graphical representation of the energies present in the HOMO and LUMO boundary orbitals of the Pentachlorophenol molecule (optimized by GaussView®software (ver.5.0).

### 2.6. Determination of Ecotoxicity of Emerging Contaminants

To quantify the levels of toxicity caused by the studied pollutants in the environment, the toxicity equivalents (TEQ) were calculated using Equations (1)–(3) [22,30,31].

$$\log \frac{1}{CE_{I50}} = 0.792 \log K_i ow - 1.735, \tag{1}$$

$$\sigma_{Pi} = \frac{\left(\sum_1^n LUMO_i\right)(\mu_i) - \left(\sum_1^n HOMO_i\right)(\mu_i)}{\left(\sum_1^n HOMOi\right)(\mu_i)}, \tag{2}$$

$$\log \frac{CE_{I50}}{\mu_i} = 24.1\sigma_{Pi} - 0.28 = (TEQ), \tag{3}$$

where $CE_{50}$: Effective concentration. *Kow*: Octanol/water coefficient. $\sigma_p$: Environmental kinetic coefficient. $\mu$: EC concentrations value.

Table 1 shows the values of the pharmacokinetic parameters for the emerging contaminants.

**Table 1.** Pharmacokinetic characteristics of contaminants.

| Contaminant | Molecular Formula | pKa | Solubility in Water (mg/L) | Bioassay | EC$_{50}$ (mg/L) |
|---|---|---|---|---|---|
| Diclofenac | C$_{14}$H$_{11}$NCl$_2$O$_2$ | 4.15 | 19.4 | *Vibrio Fischeri* 30 min<br>*Daphnia Magna* 48 h<br>*D: Subspicatus* 3 d [32] | 13.5<br>224.30<br>72 |
| Ibuprofen | C$_{13}$H$_{18}$O$_2$ | 5.2 | 21 | *Daphnia Magna* 48 h<br>*L. macrohiruz*(fish) 48 h<br>*S.costatum* 48 h [32] | 9.06<br>10<br>7.1 |
| Ketorolac | C$_{15}$H$_{13}$NO$_3$ | 3.84 | 15 | *Rat* 96 h [33] | 189 |
| PCP | C$_6$Cl$_5$OH | 4.74 | 20 | *Palemonetes pugio* 96 h<br>*Pimephalespromelas* 96 h<br>*Oncorhynchus mykiss* 96 h [34] | 0.515<br>0.19<br>0.23 |
| Estradiol | C$_{18}$H$_{24}$O$_2$ | 10.33 | 3.6 | *Fathead Minnows* 3 d<br>*Limneastagnalisa* [35] | 0.001<br>0.004 |

The index proposed by Veith and Konasewich (see in Table 2) [31] was used to classify the level of environmental toxicity of the contaminants.

**Table 2.** Veith and Konasewich index.

| TEQ | Toxicity Classification |
|---|---|
| <1 | Low or no toxicity |
| 1–10 | Toxic |
| 11–100 | Very Toxic |
| >100 | Extremely toxic |

## 3. Results

From the water samples analyses, it was possible to identify the presence of the envisioned ECs. In the next graph (Figure 8) are presented the average concentrations in the five studied effluents.

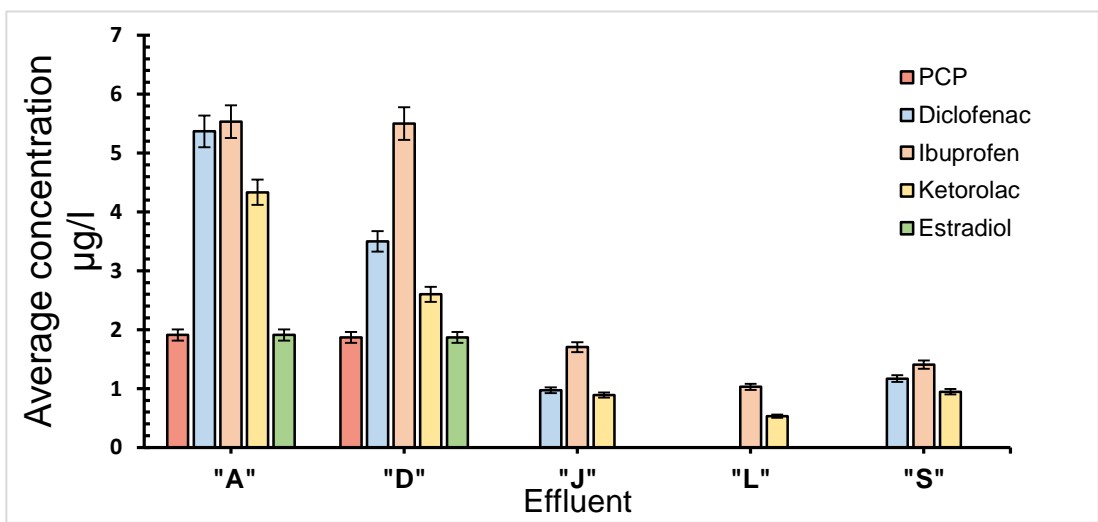

**Figure 8.** Concentration of Emerging Contaminants found in study sites.

The average values of the measured parameters at characterization for each effluent, as well as the organic loads calculated for each pollutant analyzed, are shown below in Table 3.

**Table 3.** Average characterization values of the effluents.

| Parameter | Effluent | | | | |
|---|---|---|---|---|---|
| | **A** | **D** | **J** | **L** | **S** |
| T (°C) | 28.9 ± 0.02 | 30.1 ± 0.5 | 29.9 ± 0.1 | 29.3 ± 0.1 | 31.2 ± 0.2 |
| pH | 5.9 ± 0.8 | 8.2 ± 0.2 | 6.4 ± 0.4 | 6.9 ± 0.7 | 7.56 ±0.5 |
| Conductivity (mS/cm$^3$) | 1.215 ± 0.32 | 1.448 ± 0.86 | 1.325 ± 0.65 | 1.329 ± 0.76 | 1.775 ± 0.82 |
| TDS (g/L) | 0.527 ± 0.24 | 1.987 ± 0.32 | 0.882 ± 0.12 | 0.768 ± 0.11 | 1.072 ± 0.16 |
| % Salt | 0.79 ± 0.06 | 0.67 ± 0.13 | 0.72 ± 0.56 | 0.96 ± 0.03 | 0.83 ± 0.24 |
| DO (mg/L) | 0.82 ± 0.9 | 1.94 ± 0.8 | 1.26 ± 0.6 | 0.53 ± 0.4 | 0.68 ± 0.3 |
| P (mmHg) | 759.8 ± 0.2 | 757.8 ± 0.1 | 759.6 ± 0.4 | 759.3 ± 1.1 | 758.8 ± 1.2 |
| Q (L/min) | 62.5 ± 8.2 | 3333.33 ± 26.4 | 450.5 ± 38.2 | 1562.5 ± 124.3 | 328.12 ± 19.2 |
| Diclofenac (mg/min) | 0.335 ± 0.056 | 11.66 ± 0.32 | 0.43 ± 0.072 | 0 | 0.38 ± 0.023 |
| Ibuprofen (mg/min) | 0.345 ± 0.062 | 18.33 ± 0.56 | 0.76 ± 0.11 | 1.60 ± 0.35 | 0.46 ± 0.06 |
| Ketorolac (mg/min) | 0.270 ± 0.038 | 8.66 ± 0.67 | 0.40 ± 0.08 | 0.83 ± 0.032 | 0.31 ± 0.07 |
| PCP (mg/min) | 0.119 ± 0.024 | 6.23 ± 0.23 | 0 | 0 | 0 |
| Estradiol (mg/min) | 0.033 ± 0.018 | 2.13 ± 0.12 | 0 | 0 | 0 |

The results of the ecotoxicological assessments for the identified contaminants are presented in Table 4.

**Table 4.** Ecotoxicological assessment for ECs.

| Effluent | Emerging Contaminant | | | | | | | | | |
|---|---|---|---|---|---|---|---|---|---|---|
| | Diclofenac | | Ibuprofen | | Ketorolac | | PCP | | Estradiol | |
| | **TEQ** | **Toxicity Class** | **TEQ** | **Toxicity Class** | **TEQ** | **Toxicity Class** | **TEQ** | **Toxicity Class** | **TEQ** | **Toxicity Class** |
| A | 3 | Toxic | 3 | Toxic | 3 | Toxic | 4 | Toxic | 18 | Very Toxic |
| D | 3 | Toxic | 3 | Toxic | 3 | Toxic | 4 | Toxic | 18 | Very Toxic |
| J | 3 | Toxic | 3 | Toxic | 3 | Toxic | *Nd | - | *Nd | - |
| L | *Nd | - | 2 | Toxic | 2 | Toxic | *Nd | - | *Nd | - |
| S | 3 | Toxic | 3 | Toxic | 3 | Toxic | *Nd | - | *Nd | - |

*Nd (Not detected).

According to the Veith and Konasewich index, the values of TEQ for Diclofenac, Ibuprofen and Ketorolac in all five effluents (Table 4) were categorized as "toxic".

These compounds, however, represent a greater impact to the environment because the combination of these can originate a synergic effect and potentiate their toxicity, since they have the peculiarity of being chemical species with similar properties [32,33]. It has been reported that compounds such as diclofenac in the aquatic environment affects the gill tissues and the kidneys of freshwater fish, suggesting a possible risk for this type of population [36].

Estradiol was found in concentrations of 0.61 and 0.64 µg/L in the "A" and "D" effluents respectively, (Figure 8) showing TEQ values that fall within the "very toxic" classification (Table 4). This category indicates that this compound represents a risk for aquatic biota. According to the EC$_{50}$ values presented in Table 1, this compound can probably be detected in the organisms living in the study sites, and consequently the endocrine disruption effects may be known.

In the "A" and "D"effluents the presence of the pesticide Pentachlorophenol (PCP) was confirmed (Figure 8). According to the Mexican Official Standard NOM-052 SEMARNAT-2005—"Characteristics, identification process, classification, and listing of hazardous waste", the use of this compound is restricted and it is on the list of hazardous waste within the classification of acute toxic.

The concentrations of the pesticide Pentachlorophenol in the studied effluents allow classifying it as "toxic" (Table 4). This compound suggests a greater level of impact since it reports a high value

of bioavailability [37]. This factor represents, in a numerical way, the dynamics of the compound in an organism, which can be perceived as the degree of affectation or the level of impact generated by the pollutant onto the environment.

The "L" effluent indicates the presence of the ECs Ibuprofen and Ketorolac in toxic concentrations (Figure 8). Due to the final disposal site and the permeable conditions of the soils in these areas, it is possible to speculate the infiltration of these contaminants to the subsoil and, as a consequence, the contamination of groundwater [4].

The comparison of results between the "J" and "S" effluents (Figure 8) shows that the wastewater treatment plant processes do not remove the emerging pollutants identified within the group of drugs, with the consequence of them being transported to the lower parts of coastal areas.

## 4. Conclusions and Comments

According to the performed ecotoxicological analyses, it can be concluded that the discharge of wastewater produced in the coastal zone of Cihuatlán, Jal. contain emerging contaminants in concentrations within the range of toxic. The presence of these contaminants in the coastal and estuarine ecosystems represents pollution of surface and ground water and alterations in the endocrine system of the organisms living in these environmental compartments.

The combination of pharmaceutical drugs Diclofenac, Ibuprofen and Ketorolac poses a stronger impact onto the environment. Therefore, to estimate the realistic level of ecotoxicological effect that an emerging contaminant represents, it is necessary to produce an inventory of the possible substances present in a water sample in order to study the possible synergistic or antagonistic effects that the other chemical species represent in the residual effluent.

Therefore, this study enhances the importance of carrying out additional research on the toxic effects that these type of pollutants can cause in higher organisms living in water and sediments. Further research should focus on the analyses of pollutants such as antibiotics, hormones, analgesics, and psychotropic medications, especially those that are released into the environment in large quantities and that are expected to have environmental effects, as well as focus on the study of their physicochemical characteristics.

The current study demonstrates the presence of five ECs in the Ramsar ecological study site. The wastewater treatment processes are not adequate for the removal of ECs, as products such as drugs, hormones, hygiene products, drugs of abuse, etc., can be transported within treated wastewaters in their original form and/or in their metabolized form.

Research and legislation on emerging contaminants are practically non-existent in Mexico. Hence the research summarized in this report addresses this problem through a new framework of environmental quality criteria from the response data of biological systems (chronic or acute bioassays), in addition to providing data on the environmental exposure, the destination, and the impacts generated by wastewaters in coastal zones of the country.

**Author Contributions:** J.A.M.-P. and A.T.-G., conceived and directed the project M.Á.A.-P. performed part of the experiments and wrote the paper; E.R.-A., E.G.-D. and F.d.A.S.-B. further improved the concept, structure, contents and writing of the manuscript and also contributed to the discussion.

**Funding:** This research received no external funding.

**Acknowledgments:** JAMP, ATG, ERA, EGD, and FSB wishes to thank Instituto Politécnico Nacional (SNI-CONACyT, EDI, and COFAA), México, for their support. The present study was mainly funded by SIP-IPN, Mexico (No. 20180081 and 20190101). MAAP thanks CONACyT, Mexico, for the research fellowship. Special thanks to Javier Agustín Flores.

**Conflicts of Interest:** The authors declare no conflict of interest. The founding sponsors had no role in the design of the study; in the collection, analyses, or interpretation of data; in the writing of the manuscript, and in the decision to publish the results.

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
