# Peer review of "Ecotoxicological Analysis of Emerging Contaminants from Wastewater Discharges in the Coastal Zone of Cihuatlán (Jalisco, Mexico)"

_water, doi:10.3390/w11071386_

Round 1
Reviewer 1 Report
This paper assesses the ecotoxicity of 5 effluent wastewater samples collected in a certain location in Mexico regarding the presence of 5 emerging contaminants, namely, Diclofenac, Ibuprofen, Ketorolac, Pentachlorophenol (PCP), and Estradiol.
The ecotoxicity is evaluated based on the calculation of a toxicity equivalent (TEQ), which should be a function of the toxicity to a given microorganism, and the compound concentration in the samples.
It is not clear from the three equations presented, how the TEQ was calculated. From Eq. (1), Log (1/CEI50) can be calculated as a function of the compound Kow, but then Log (CEI50) would be known.
How are the ECs concentrations included in the TEQ? In line 209, the Greek letter mu has a different meaning.
Specific comments:
Abstract, lines 26 – 29 “This research recognizes that the analysis of the physicochemical properties of substances allows predicting whether a contaminant is likely to act and persist in the environment and, in turn, bioaccumulate in organisms, in addition to estimating the possible synergistic and antagonistic effects between the chemical species of the pollutants.”
Of course the chemical properties of a substance determine their behaviour in the environment, however, their synergistic and antagonistic effects are much more difficult to estimate, and I don’t understand how this particular study contributes to estimating these effects (synergistic and antagonistic). Therefore I would recommend deleting this sentence.
Line 56 “…and a null or limited availability of methods for their analysis [6].”
Although a reference is given I cannot agree to this statement, as there are thousands of papers dedicated to the analysis of pharmaceuticals and pesticides in the different environmental compartments, particularly, the aquatic environment.
Lines 64-66 Not all drugs, resins, plastics, pesticides, detergents, cosmetics, fragrances, etc. are organic disruptor compounds.
Lines 81 – 82 “Currently, the presence and destination of emerging active pollutants in the aquatic environment is one of the priority events in environmental chemistry [7]”
“…is one of the priority topics in environmental chemistry.”
Lines 82-84 The aspects listed are only part of the whole problem. Indeed they are important, but besides monitoring and treatment, also the toxicity assessment is also very important.
Lines 97 – 98 The definition of the toxicity parameters is incomplete.
Lines 111-112 “…likewise, it shows that the physicochemical properties of pollutants are conditioned by their distribution, bioaccumulation, and biomagnification.”
It is quite the contrary, “…the distribution, bioaccumulation, and biomagnification of these pollutants is conditioned by their physicochemical properties.”
Line 141 2.2 Water sample extract spiking
The section title is not in agreement with the information given
I would recommend presenting all the figures relating to the effluent location in 1 or 2 pages only.
Lines 191 -194 I’m not used to encounter this type of extraction with dichloromethane for these compounds; it’s not usual to clean-up dichloromethane extracts in C18 SPE columns. The authors should provide a reference for the extraction method used.
Table 3 Replace “sal” by “salt” if appropriate.
Lines 293-296 I don’t think it’s clear how the results from the bioassays were incorporated in this study.
As a conclusion, and although recognizing the value of this type of studies I do not recommend its publication in the present form.
Author Response
Dear reviewer
Thanks for the observations made.
The manuscript was improved taking into account all the suggestions and specific comments. The file is attached with the new version, for your consideration.
Kind regards
Specific comments:
Abstract, lines 26 – 29“This research recognizes that the analysis of the physicochemical properties of substances allows predicting whether a contaminant is likely to act and persist in the environment and, in turn, bioaccumulate in organisms, in addition to estimating the possible synergistic and antagonistic effects between the chemical species of the pollutants.”
Response: It was changed as recommended
Line 56“…and a null or limited availability of methods for their analysis [6].”Lines 64-66 Not all drugs, resins, plastics, pesticides, detergents, cosmetics, fragrances, etc. are organic disruptor compounds. Lines 81 – 82“Currently, the presence and destination of emerging active pollutants in the aquatic environment is one of the priority events in environmental chemistry [7]”Lines 82-84 The aspects listed are only part of the whole problem. Indeed they are important, but besides monitoring and treatment, also the toxicity assessment is also very important. Lines 97 – 98 The definition of the toxicity parameters is incomplete. Lines 111-112 likewise, it shows that the physicochemical properties of pollutants are conditioned by their distribution, bioaccumulation, and biomagnification.”It is quite the contrary, “…the distribution, bioaccumulation, and biomagnification of these pollutants is conditioned by their physicochemical properties.”
Response: It was changed and improvement as recommended
Line 141 2.2 Water sample extract spiking. The section title is not in agreement with the information given
Response: It was changed
I would recommend presenting all the figures relating to the effluent location in 1 or 2 pages only.
Response: It was changed as recommended
Lines 191 -194 I’m not used to encounter this type of extraction with dichloromethane for these compounds; it’s not usual to clean-up dichloromethane extracts in C18 SPE columns. The authors should provide a reference for the extraction method used.
Response: The reference was included
Table 3 Replace “sal” by “salt” if appropriate.
Response: It was changed as recommended
It is not clear from the three equations presented, how the TEQ was calculated. From Eq. (1), Log (1/CEI50) can be calculated as a function of the compound Kow, but then Log (CEI50) would be known. How are the ECs concentrations included in the TEQ?
Response: Was corrected
Reviewer 2 Report
Manuscript ID: water-505883
Title: Ecotoxicological analysis of emerging contaminants from wastewater discharges in the coastal zone of Cihuatlán (Jalisco, México)
This present research works deals with the examination of emerging contaminants from wastewater discharges in the coastal zone of Cihuatlán (Jalisco, México). The author selected actual wastewater sources specifically produced by urban and rural settlements of the coastal zone of Cihuatlán. To investigate the detect emerging contaminants (Diclofenac, Ibuprofen, Ketorolac, Pentachlorophenol (PCP), & Estradiol) concentration. The authors successfully confirmed the presence of the EC’s in all the selected sites. Authors confirmed that the concentrations of pollutants Diclofenac, Ibuprofen, Ketorolac, and Pentachlorophenol are within the toxic range, whereas the compound Estradiol was found in concentrations that represent high toxicity. Overall the present study brings out some interesting research related to the field of ecotoxicological analysis. Might be published results will be suitable for the awareness of the science/society and the wastewater treatment researchers. I suggest this paper to be accepted suitable for publication in WATER.
Author Response
Dear reviewer
Thanks for the observations made.
The manuscript was improved taking into account all the suggestions and specific comments. The file is attached with the new version, for your consideration.
Kind regards
Round 2
Reviewer 1 Report
This is the revised version of a paper of which I have commented the first version. The authors have made some changes and corrections to the manuscript, which have generally improve the quality of the manuscript.
Nevertheless, two points still need to be addressed by the authors:
1. I haven’t seen any change to the abstract regarding my comment on how this study allows to predict synergistic and antagonistic effects between these compounds, and I believe this cannot be assessed by this type of study.
2. Lines 204 -207. When I recommended the addition of a reference regarding the extraction methodology, I was expecting that a paper describing this method and the validation parameters for these analytes was presented, and not a review article, which reviews microextraction techniques, and this was not used. As described by the authors liquid-liquid extraction with dichloromethane followed by SPE C18 clean-up was used for analytes' extraction.
Ref. [28] “In this work, we review the most recent applications of microextraction preparation techniques in different water environmental matrices to determine organic micropollutants: solid-phase microextraction SPME, in-tube solid-phase microextraction (IT-SPME), stir bar sorptive extraction (SBSE) and liquid-phase microextraction (LPME).”
Line 77 “sufamethoxazole” should be changed to “sulfamethoxazole”
If I’m not mistaken, references 38 and 39 are not cited in the text.
Therefore, and although recognizing the value of this type of studies and the improvements to manuscript made by the authors, I do not recommend its publication in the present form.
Author Response
Dear reviewer
Thanks for the observations made.
The manuscript was improved taking into account all the suggestions and specific comments. The file is attached with the new version, for your consideration.
Kind regards
Specific comments:
1. I haven’t seen any change to the abstract regarding my comment on how this study allows to predict synergistic and antagonistic effects between these compounds.
Response: It was changed as recommended
2. Lines 204 -207. When I recommended the addition of a reference regarding the extraction methodology, I was expecting that a paper describing this method and the validation parameters for these analytes was presented As described by the authors liquid-liquid extraction with dichloromethane. Ref. [28]
Response: It was changed
“sufamethoxazole” should be changed to “sulfamethoxazole”
Response: Was corrected
references 38 and 39 are not cited in the text.
Response: Was corrected.
Round 3
Reviewer 1 Report
This is the revised version of a paper of which I have commented the first and second versions. The authors have made changes and corrections to the manuscript, which have generally improved the quality of the manuscript.
I believe that the manuscript is now suitable for publication.